# Graph-based pangenomics maximizes genotyping density and reveals structural impacts on fungal resistance in melon

Justin N. Vaughn [1,2] ✉, Sandra E. Branham[3], Brian Abernathy[2], Amanda M. Hulse-Kemp[4,5], Adam R. Rivers [6], Amnon Levi[7] & William P. Wechter[3,7] ✉

The genomic sequences segregating in experimental populations are often highly divergent from the community reference and from one another. Such divergence is problematic under various short-read-based genotyping strategies. In addition, large structural differences are often invisible despite being strong candidates for causal variation. These issues are exacerbated in specialty crop breeding programs with fewer, lower-quality sequence resources. Here, we examine the benefits of complete genomic information, based on long-read assemblies, in a biparental mapping experiment segregating at numerous disease resistance loci in the non-model crop, melon (*Cucumis melo*). We find that a graph-based approach, which uses both parental genomes, results in 19% more variants callable across the population and raw allele calls with a 2 to 3-fold error-rate reduction, even relative to single reference approaches using a parent genome. We show that structural variation has played a substantial role in shaping two *Fusarium* wilt resistance loci with known causal genes. We also report on the genetics of powdery mildew resistance, where copy number variation and local recombination suppression are directly interpretable via parental genome alignments. Benefits observed, even in this low-resolution biparental experiment, will inevitably be amplified in more complex populations.

Reference-quality genomes can now be generated on a study-specific basis thanks to recent improvements in cost and quality of long-read sequencing. In the simplest scenario, a geneticist may want to have genomes available from both parents for genotyping a biparental mapping population. For a conventional recombinant inbred line (RIL) population, this approach will reveal all genetic variants segregating in the population and create a complete reference for founder haplotypes. Such data will inevitably aid prioritization of candidate genes within an associated interval, particularly when presence/absence variation plays a role. Founder assemblies will also increase the number of properly aligned reads from progeny samples and improve the accuracy of mapping quality scores for downstream filtering[1]. One major objective of this study is to look at the degree to which these theoretical advantages are

[1]Genomics and Bioinformatics Research Unit, The Agricultural Research Service of U.S. Department of Agriculture, Athens, GA 37605, USA. [2]Department of Crop and Soil Sciences, University of Georgia, Athens, GA 30602, USA. [3]Plant and Environmental Sciences Department, Coastal Research and Education Center, Clemson University, Charleston, SC 29414, USA. [4]Genomics and Bioinformatics Research Unit, The Agricultural Research Service of U.S. Department of Agriculture, Raleigh, NC 27965, USA. [5]Department of Crop and Soil Sciences, North Carolina State University, Raleigh, NC 27695, USA. [6]Genomics and Bioinformatics Research Unit, The Agricultural Research Service of U.S. Department of Agriculture, Gainesville, FL 32608, USA. [7]US Vegetable Laboratory, The Agricultural Research Service of U.S. Department of Agriculture, Charleston, SC 29414, USA. ✉e-mail: justin.vaughn@usda.gov; pat.wechter@usda.gov

reflected in genotyping accuracy and, finally, genetic resolution and interpretation.

With long-read assemblies of two parents in hand, ideally both could be used as references for genotyping to avoid alignment bias and to capture all variation. Some attempts have been made to address this issue in a single-reference paradigm by using a consensus approach that includes all relative insertions[2]. While the consensus makes read alignment more robust, the approach is still susceptible to breakpoint misinterpretation and does not take full advantage of known small-scale differences—single-nucleotide polymorphisms and indels—when aligning reads. Graph-based approaches have emerged as a way to address all variation, particularly structural variants (SVs) greater than 50 bp and divergent repeats[3]. The method represents a theoretically complete solution to reference bias but also a very different bioinformatic philosophy and toolkit. Hybrid methods, such as Practical Haplotype Graphs[4], align reads to a single reference, but then use this information to impute the path through a graph of gene-centric reference ranges. Ideally, full genomic information could be used during the read alignment step such that imputation and visualization are as accurate as possible.

Of bioinformatic programs built within the graph framework, the vg-toolkit currently offers the most exhaustive end-to-end, open-source solution for graph-enabled genetic analysis[5]. Now part of the vg toolkit, the giraffe aligner has been shown to reduce positional alignment error in maximum map quality (MQ) reads from one mismapping in 3333 (using bwa-mem) to one in 142,857 while maintaining total read alignment values comparable to the best single-reference aligners[1]. Since these results were based on relatively homogenous human genomes, we expect the technique, if scalable, to be even more useful in structurally divergent crop genomes.

Reduced representation sequencing (RRS) and skim-sequencing (skim-seq) are two major sequence-based genotyping approaches used to characterize mapping populations[6]. As actual base-call sequencing costs fall and library prep costs achieve parity, skim-seq has become a more frequent choice due to its conceptual simplicity and unbiased coverage. Heterozygous calls remain a challenge in skim-seq because they cannot be called directly due to limited read depth. Imputation offers a way to address heterozygosity, miscalls, and missing data common to skim-seq (and RRS), although this depends on the accuracy of variant identification. Imputation is usually supported by much deeper read coverage of parents. In a conventional single-reference paradigm, variant accuracy is a function of the divergence between the reference and the parents. Such divergence often involves, for example, tens of thousands of SVs in a rice breeding cross[7] and hundreds of thousands—if not millions—in a maize experimental cross[8]. In this study, we were interested in what comparable or contrasting benefits, if any, graphs built with de novo assembled parental genomes would have over these standard genotyping approaches.

Disease-resistance loci are frequent sites of rampant structural variation across all kingdoms of life. In plants, nucleotide-binding site leucine-rich repeat (NBS-LRR) genes have direct interaction with fungal effectors, are often clustered within the genome, and have undergone substantial phylogenetic divergence[9,10]. In addition, this variation is often rare in the population because local factors critical for disease manifestation result in local adaption of the pest and its host. This combination of factors makes disease resistance an ideal test case to explore the benefits of a graph-based approach to genetic analysis, particularly in the context of structured populations. *Fusarium oxysporum* f. sp. *melonis* race-1 resistance has been fine mapped using the population in this study to the *fom-2* locus on chromosome (Chr) 11, which contains an NBS-LRR protein (encoded by *MELO3C021831*) characteristic of resistance across plant species[11,12]. Similarly, resistance to race-2 of the same fungal disease has also been traced to a distinct NBS-LRR gene (*fom-1; MELO3C022146*) on Chr 09[13,14].

In this work, we develop a computational pipeline for graph-enabled, low-coverage genotyping and genetic mapping. We also contrast this approach with conventional methods, revealing a ~19% increase in useful variants. We then focus these methods on two critical agronomic traits in melon—*Fusarium* and powdery mildew resistance (see above). In many of these cases, association analysis and comparative genomics indicates causal variation is likely related to large, structural mutations that have previously been recalcitrant to short-read assembly and single-reference genotyping.

## Results

### Parental chromosomes are highly collinear but reveal extensive, fine-scale structural differences

AY and MR1 parental genomes were sequenced using highly accurate circular consensus sequencing of long-reads on the PacBio Sequel IIe. Resulting assemblies had contigs with an N50 of 9.9 and 9.1 Mb for AY and MR1, respectively (Table 1). Contigs of each sample were initially scaffolded using the community reference, DHL92, as well as reciprocally (see "Methods") into pseudomolecules averaging 30 Mb (Table 1). Dotplots of pseudomolecules reveal high collinearity for all chromosomes except Chr 06, which, though the majority of homologous sequence is present, ordering is highly variable across all three genomes. Our pangenomic pipeline is robust to large-scale chromosomal variation if major inter-chromosomal translocations are not present, so we did not interrogate these inversion/translocations further.

Chromosomes were annotated using an augmented BRAKER2 pipeline. Both Curcurbit protein homologs and previously published RNA-seq data from various sources including MR1 and AY were combined to create gene annotations. Genomes were annotated separately. We observed that conventional repeat modeler approaches to masking were overly aggressive and were disrupting accurate exon annotation. As an alternative we used a k-mer based approach (see "Methods"). Thus, our final gene sets likely harbor some low-copy transposable element proteins. We also produced repeat annotations such that they could be overlaid with the final gene set to assess this possibility on a case-by-case basis.

The community reference genome for melon, based on variety DHL92, has undergone numerous updates. Though a long-read assembly was recently published[15], we focus on version 3.6.1 in what follows because it was used in cuGenDB (http://cucurbitgenomics.org/) at the time of writing and illustrates issues related to short-read assemblies. (Version 2 of cuGenDB was released in February 2022 and contains all genome versions.) Multi-sequence alignments of MR1, AY, and DHL92 were generated on a per-chromosome basis. Across all chromosomes, 287.0 Mb (~75% of MR1 or AY, see below) is shared by all three genomes. The large-scale collinearity of these three genomes (Supplementary Figs. 1–12) tends to hide the substantial degree of unique sequence being contributed by each sample (Fig. 1). Such sequence is either non-orthologous or is too divergent to align

**Table 1 | Assembly statistics for parental genomes and community reference DHL92 (v3.6.1)**

|  | AY | MR1 | DHL92 |
|---|---|---|---|
| Total number of bases (bp) | 20,084,631,307 | 25,786,193,521 | NA |
| Mean read length (bp) | 12,228 | 11,377 | NA |
| Contig # | 2301 | 3729 | 42,067 |
| Contig total size (Mb) | 453.64 | 488.058 | 337.33 |
| Contig N50 (Mb) | 9.885 | 9.072 | 0.262 |
| Scaffold N50 (Mb) | 30.46 | 29.52 | 34.32 |
| BUSCO-complete-embryophyta_odb10 | 1579 | 1579 | 1570 |
| BUSCO-complete % | 97.8% | 97.8% | 97.3% |

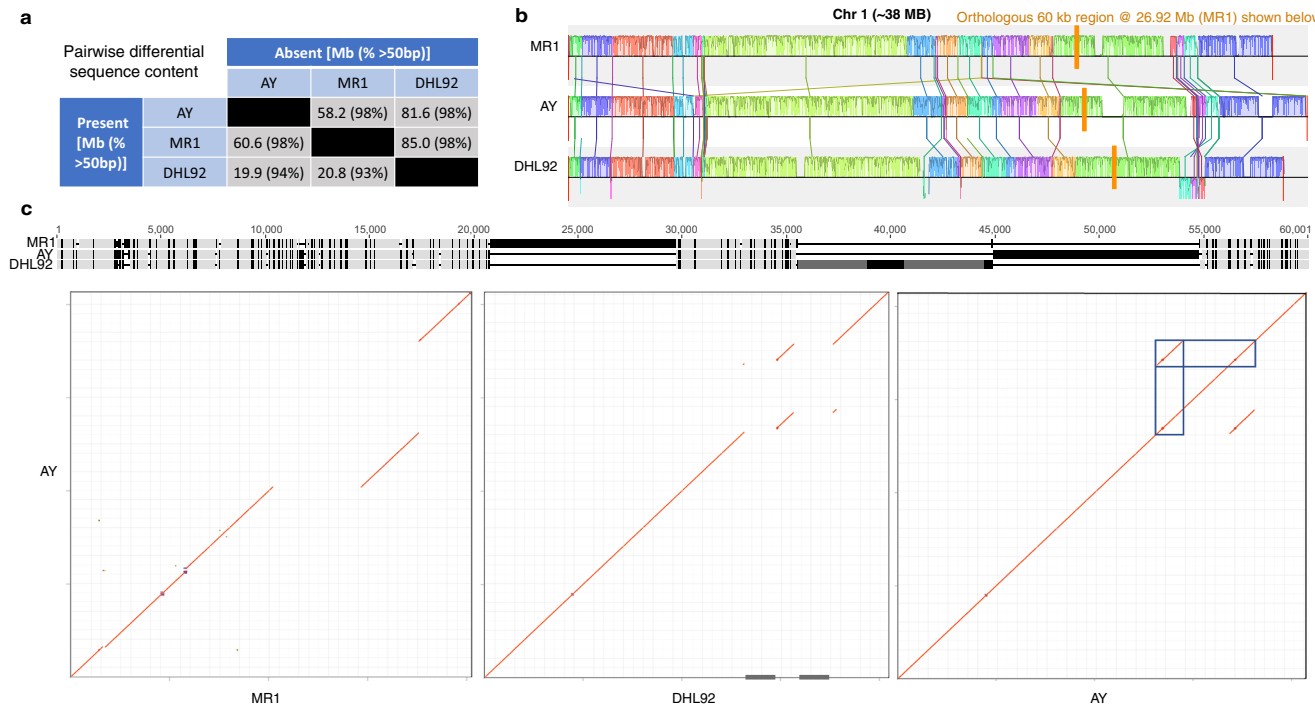

**Fig. 1 | Pangenome of three melon sequences and sources of genomic sequence. a** Relative presence/absence of genome sequence is depicted for pairwise comparison (based on multi-sequence alignments). Only variants >50 bp are shown but their percentage of the total is indicated parenthetically. **b** Chromosome-scale MSA of Chr01. Sequences do not contain gaps and so are spatially offset. See mauve viewer for details (https://darlinglab.org/mauve/user-guide/viewer.html). Colors reflect locally collinear blocks. Rare, relative inversions appear below the forward strand sequence. The orthologous region shown in **c** is highlighted in orange. **c** A 60 kb sample from Chr01 MSA highlighted in **b** is shown with light gray indicating identical columns and black representing variable columns. Dark gray regions in DHL92 are "N"s. The lower panel is a dotplot representation of this region with AY along the *Y*-axis of each. Red indicates an identical region and blue, a highly similar match. The last plot is AY by self. Blue boxes highlight the tandemly repetitive nature of the AY insertion. Note this repeat region falls over the long Ns tract in DHL92.

at single-nucleotide resolution. The original DHL92 genome was generated from relatively short 454 reads and so is expected to be far from complete. Indeed, relative to DHL92, AY and MR1 possess 84.6 Mb of additional sequence on average whereas DHL92 only adds 21.8 Mb on average (Fig. 1a and Supplementary Data 1). Relative to one another, AY and MR1 add 60.4 Mb on average not accounted for in the other genome. In terms of total DNA content, 98% of the differences between MR1 and AY are caused by SVs (>50 bp). That said, short indels (<50 bp) occur ~14-fold more often (Supplementary Data 1). As expected, the 2,106,962 single-nucleotide variants between AY and MR1 comprise the majority (81%) of differentiating variants, although they only affect a miniscule proportion of altered bases (2%) due to the substantial size of many SVs. A three-way analysis based on unique nodes in the graph (which does not differentiate variant type) was complementary and reveals MR1 and AY contribute ~55 Mb of primary non-orthologous sequence on average (Supplementary Fig. 13). Given the degree of unique sequence that each genome brings to the alignment, we next contrasted genotyping quality between single-reference methods missing this information and graph-based methods that include it.

## Graph-enabled alignment increases genotype-able variants and reduces error rate of allele calls

All chromosome alignments were merged and used to construct a pangenomic graph (Fig. 2). All three genomes could be completely reconstituted as continuous, acyclic paths through this graph. A population of 149 RILs derived from the MR1xAY cross were skim sequenced at ~1x coverage as 150 bp paired-end reads. To contrast distinct library approaches, we also used pre-existing genotype-by-sequencing (GBS) 100 bp single-end reads derived from the same DNA extractions[12]. All reads were then aligned to the graph using giraffe. This aligner implements a situational strategy that will use all

haplotype combinations implied by the graph in variant-sparse regions but, to avoid combinatorial explosion, will revert to the foundational paths in variant-dense regions. We contrasted this graph construction and alignment approach, called PanPipes hereafter, with a conventional single-reference strategy (see "Methods") using two different references: DHL92, the community reference, and MR1, the donor parent for all major resistance alleles in the cross. We refer to these as single-reference-DHL92 (SR-DHL92) and single-reference-MR1 (SR-MR1) in what follows.

Given the divergence in sequence content seen above, it might be expected that far more reads would align in the PanPipes case because AY insertions (relative to MR1) were present to capture reads that would otherwise not have a target. We observed the opposite (Table 2): across samples, 3% fewer reads on average were aligned by giraffe than by bwa-mem against MR1. Filtering out reads with MQ < 50 further increased this difference to 8%. We manually examined reads that were aligned by bwa-mem but not by giraffe and vice-versus. Two aspects account for differing read content: (1) giraffe is far more stringent and frequently rejects reads that have soft-clipping or are hyper-variable in bwa-mem. Generally, this behavior is desired, particularly when all variants should be present in the graph. (2) For efficiency, giraffe will ignore a read if there are too many seed matches. Though problematic for interpretation, the approach is, in effect, an MQ-filter built into the alignment stage. Because we use an explicit MQ filter, this optimization does not affect our results, but it could be an issue for researchers attempting to genotype variants for which one allele is highly repetitive. Taken together, much of the non-orthologous sequence observed above is clearly a product of >150 bp repeats.

An additional striking and unintuitive result is that aligning to the less complete DHL92 genome results in 9% and 17% more MQ > 50 alignments than using MR1 alone or both parents (PanPipes), respectively (Table 2). This observation points to an under-explored aspect of

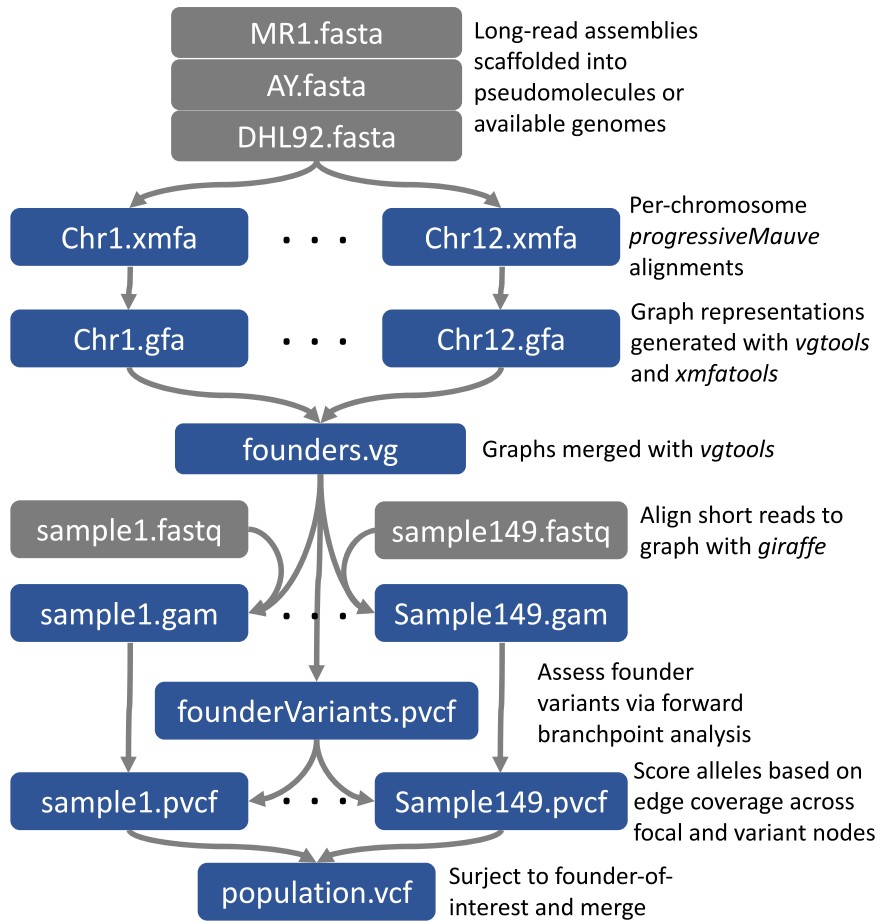

**Fig. 2 | PanPipes flow diagram.** PanPipes incorporates pangenome alignment information during short-read mapping. Files are shown as blocks and each step is annotated on the right. Gray indicates that files originate from outside the pipeline.

"Surject" means to transform graph coordinates into coordinates of one of its constituent paths.

alignment to a divergent and/or incomplete reference: the inability to accurately assess repetitiveness. Reads that are clearly derived from duplicated loci are considered high quality in SR-DHL92. (The same effect likely explains some of the difference between PanPipes and SR-MR1, where duplicated loci in AY but not MR1 will not be filtered in the single-reference approach.) Such reads are far more likely to trigger false variants, as explored below.

The vg call function is a pre-existing tool for graph-based genotyping[5]. We found that vg call had an excessively high false positive rate because it is not designed for low-coverage data and has not been extensively tested using giraffe alignments. Instead, we incorporated a simple variant caller and genotyper directly into Pan-Pipes (Fig. 2). Much like vg call and GATK's HaplotypeCaller, the caller identifies branchpoints in the graph and calls alleles by following the path with the most coverage at these branchpoints (see "Methods").

We rely on the giraffe alignment step to assess haplotype likelihoods and, afterward, use majority rule to call the allele, which is sufficient for low-coverage RIL samples where imputation is expected to be used. This approach also allowed us to more easily trace possible genotyping errors back to alignments.

Both SR-DHL92 and PanPipes initially contain variants unique to DHL92 as well as those distinguishing AY from MR1. Thus, the initial PanPipes variant set falls from 3,813,582 to 2,261,056 after filtering out variants that are not segregating (<0.2 MAF) in the RIL population.

The SR approaches allow variants to be called from population reads while the PanPipes approach establishes a pre-ascertained set based solely on founder assemblies. This difference makes initial variant numbers for SR-MR1 and SR-DHL92 more difficult to evaluate because low-coverage data triggers numerous false variants from sequencing errors. Therefore, we considered the starting set of variants to be those present after segregation filtering (Fig. 3a).

We also removed variants that were multi-allelic or non-polymorphic in parents (Fig. 3a). Missing calls in parents were tolerated because they exhibited segregation in the population and their call was not in direct conflict. Removal of these variants had a small impact on both PanPipes and SR-MR1 relative to SR-DHL92, where calls appear much more error-prone given the divergence of the reference from the material being genotyped.

False or poorly genotyped variants will typically exhibit segregation patterns distinct from the true variants with which they should be in linkage based on the genome sequence. This aberrant pattern is particularly distinctive in experimental populations. Using a 20-marker wide window, we removed any focal variant that did not have a

**Table 2 | Percent of aligned skim-seq reads across all RIL samples using assorted methods**

| | Properly-paired + singletons | Properly-paired + singletons w/ MQ > 50 |
|---|---|---|
| SR-DHL92 (bwa-mem) | 89% | 69% |
| SR-MR1 (bwa-mem) | 98% | 61% |
| PanPipes (giraffe) | 95% | 53% |

Total reads = 776,180,980.
*SR* single reference, *MQ* mapping quality.

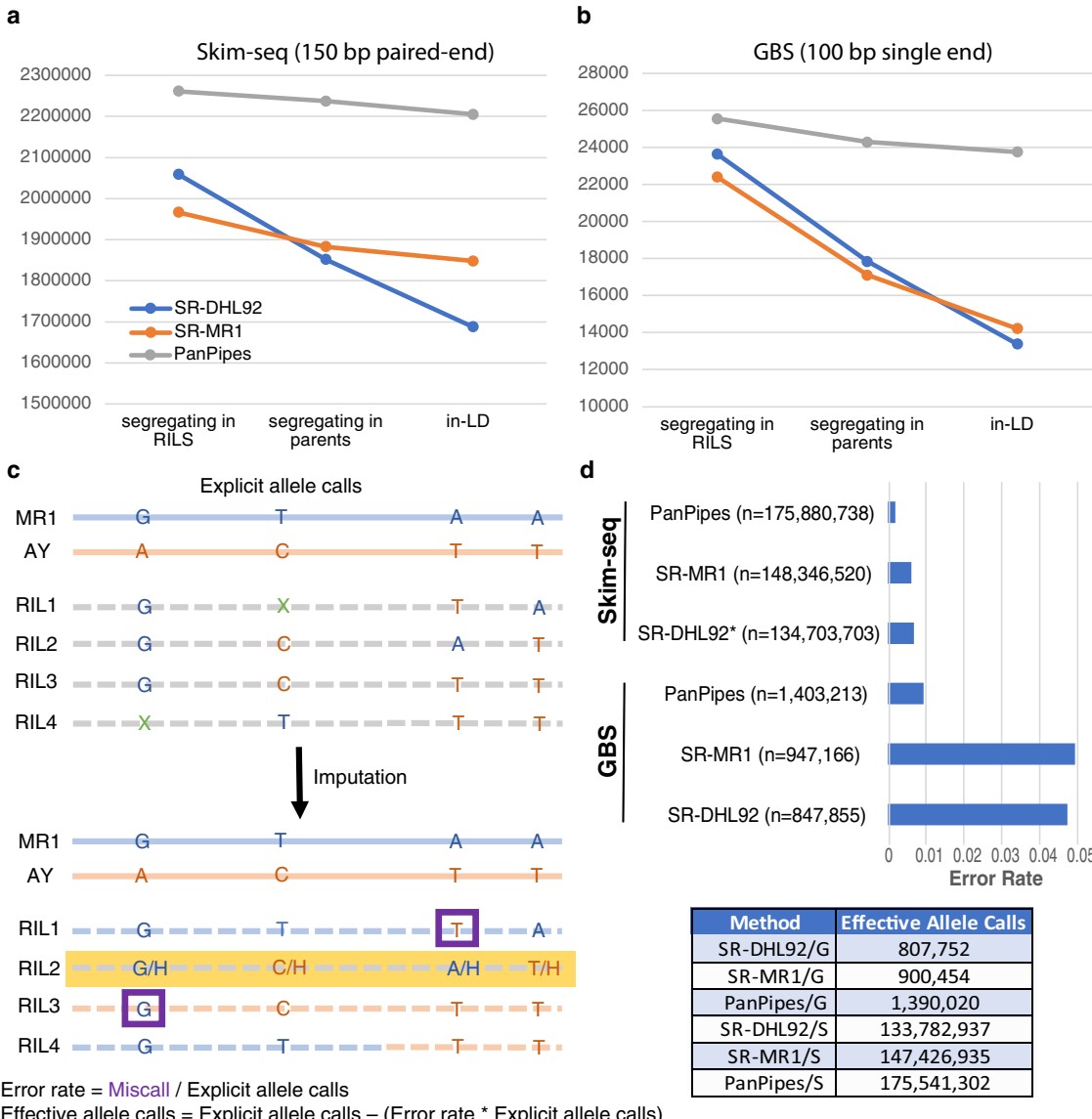

**Fig. 3 | Genotyping efficacy of biparental population across methods.** Three methods used were (1) a graph-based approach using the pangenome−PanPipes, (2) single-reference approach using newly developed MR1 assembly−SR-MR1, and (3) a single-reference approach using currently available DHL92 assembly−SR-DHL92. **a** For skim-seq chemistry, the three methods are compared in terms of starting variants and variants remaining after various filtering criteria described in text. **b** Comparable to **a** but for GBS chemistry. **c** Error rate assessment is illustrated in a toy example. Green Xs indicate missing calls. Purple boxes indicate post-imputation assessment of miscalled alleles. Yellow highlighting indicates a likely contaminant haplotype not found in either parent. **d** Box plots depict relative error rates. Adjacent table shows grand totals for effective allele calls across chemistries and methods, where G GBS, S Skim-seq. *n* values are total explicit allele calls (see **c**). Source data are provided as a Source Data file.

pairwise *D'* value >0.93 when averaged across its 19 neighbors. Like other filtering steps, these errors were far-more frequent in SR-DHL92 and, to a lesser extent, SR-MR1 (Fig. 3a).

Comparable results are observed for GBS data (Fig. 3b). The loss of specificity related to unpaired, shorter reads has a more detrimental effect on both SR approaches relative to PanPipes. Because it is based on a window of variants, linkage disequilibrium (LD) results will be confounded with variant density, so it is likely that some valid variants are being removed from genomic regions with low variant-density to recombination ratio. Still, the sharp declines after polymorphic parent removal suggest graph-based genotyping is even more valuable when unpaired or short-reads are used and that repetitive structures larger than 100 bp are driving a sizeable fraction of false variants.

Low-coverage sequencing presents additional challenges that cannot be easily detected with filters used above on a per-variant basis. Sequence errors that coincidentally match the wrong allele will result

in a miscall. More problematic, tandem and local duplications are a common feature of plant genomes[7,16]. When the reference sequence is single-copy, variants within these duplications cannot be easily filtered because they are physically proximal and will only present as three of four possible haplotypes in terms of LD calculation. Graph-based alignments should be more robust to these errors.

We used pedigree information to test the ability of each approach to call individual alleles (Fig. 3c). Genomic regions in a structured population are expected to partition into parental haplotypes. Error rates can be determined by assigning these haplotypes to all samples and cross-checking with the expected bases. Though theoretically simple, low-coverage and experimental realities present complications. The parents used in large plant populations are often technically multiple individuals from a family that has been genetically homogenized via inbreeding. Small regions that remain heterogenous across these parents will segregate in the population as regions with more

than two parental alleles. With skim-seq, heterozygosity is difficult to differentiate from this parental heterogeneity. FSFHap can assess heterozygosity and non-parental haplotypes by using an explicitly supplied pedigree model and by considering the entire population, not just parents, in initial haplotype definition steps[17]. Only regions that were explicitly called and polymorphic in parents, after imputation, were considered; thus, regions that were recalcitrant to imputation were not considered in assessing error rate.

For skim-seq, the PanPipes approach improved allele call error rates compared with a conventional single-reference approach by 3.3-fold and 5.3-fold relative to SR-MR1 and SR-DHL92, respectively (Fig. 3d). Comparable gains are seen when using GBS data, although, as seen in earlier filtering steps, the error rate is much higher for the shorter, unpaired reads. We observed that, in some cases, FSFHap imputed non-parental haplotypes as heterozygotes in PanPipes but as explicit non-parental haplotypes in SR approaches, possibly due to different variant densities. Concerned this difference might bias error rate calculations, we also developed a simple window-based imputer that leaves such regions unimputed if the heterozygosity for that region exceeds expectation. For these $F_{6-11}$ RILs, we used a 4% cross-population heterozygosity threshold. While this approach ignores highly problematic regions, it still showed substantial improvements of 3.2-fold and 4.5-fold reduction in PanPipes errors relative to SR-MR1 and SR-DHL92, respectively, for skim-seq.

For skim-seq, after major filtering steps and adjustments for error rate, the PanPipes approach calls 16% and 31% more alleles than SR-MR1 and SR-DHL92, respectively (Fig. 3d). These values are driven by the number of filtered variants in PanPipes (Fig. 3a), not by the average number of allele calls per variant: both SR-MR1 and PanPipes call, on average, ~80 out of 147 RILs (when counted as haploid).

Variant density difference between PanPipes and SR-MR1 shows an evenly distributed bias toward PanPipes across the genome and, on average, results in 5.9 more variants per 5 kb in this RIL population (Supplementary Fig. 14). The few windows for which variant density is biased toward SR-MR1 involve variants being called within a large MR1 insertion (relative to AY). Frequently, as discussed above and below, these calls involve large tandem duplications, which are difficult to identify via LD and segregation behavior. Manual curation indicates that PanPipes-biased windows are generally the result of more effective calling around highly polymorphic regions interspersed with large indels, which disrupt bwa-mem alignments but can be aligned to the graph because all variants are present.

**Pangenomic perspective reveals full variant profile of *Fusarium* race-1 wilt resistance gene**

Using known causal genes as a benchmark, we examined the impact of increased variant resolution and genotyping accuracy on genetic resolution. A basic calculation of expected recombination bin size in our population would predict that, particularly for skim-seq, coverage far exceeds that which is theoretically required to resolve all recombination blocks. We hoped exploration in this simplified biparental context would give insight into how the methods would behave across much more complex populations, where higher density variant information would be critical. To that end, for genetic mapping, we employed a mixed-effects, linear modeling approach commonly used in genome-wide association scans. This approach would be problematic with low marker density, but, in our case, a physical sequence of both parents and thus effectively all variants are available.

All association profiles around *fom2* reveal a peak anchored on the NBS-LRR gene (*MELO3CO21831*) previously implicated in Fusarium wilt race-1 resistance (Fig. 4 and Introduction). Imputation has a dramatic effect on clarifying the associations across all techniques. At higher resolution, the beneficial impact of higher density skim-seq (relative to GBS) becomes evident in the interpretation of causal

recombination bins. Imputation also clarifies the high-resolution perspective as well. Imputation proved problematic when using DHL92 as a reference with skim-seq, so we could not compare utility of a single parent reference (MR1) relative to DHL92 in this case. GBS profiles for raw genotypes are quite distinct in SR-DHL92 but become much more similar across techniques after imputations. When comparing skim-seq profiles, many regions problematic for SR-MR1 have an association signal in PanPipes more in keeping with expected recombination patterns.

A major benefit of using high-quality parent references is that trait-associated regions can be rapidly assessed in terms of their true variation, not just those implied by short-reads. Moreover, these genomes can be reannotated with confidence that all variation is factored into gene models. The genomes of MR1 and AY exhibited very similar versions of the NBS-LRR thought to underlie resistance (Fig. 5a). In fact, our re-annotations suggested that this gene is a single, continuous open reading frame (ORF), yet still exhibits little variation to warrant such dramatically different disease responses: AY, death; MR-1, no visible symptoms. We tracked the difference in exon annotations to the use of inappropriate repeat models of conventional pipelines, which frequently masked exonic space that was then interpreted as an intron. We resolve over-masking by instead employing k-mer masking based on raw reads and single-copy coverage estimates (see "Methods").

We also adopted recent advances in protein structure prediction to explore the validity of our single-exon models as well as any structural difference implied by linear sequence divergence. Six distinct isoforms of the NBS-LRR were compared: both continuous and three-exon models across MR1, AY, and DHL92 sequences. (In addition, we both randomly shuffled and reversed sequences to assure that the folding algorithm produced disorganized structures in those cases.) Both three-exon and continuous ORF models produced expected NBS-LRR structures (Fig. 5b). We cannot rule out that both isoforms are produced, but pre-existing RNA-seq data supports the continuous ORF as the primary protein product.

We then compared AY and MR1 continuous ORF proteins from a structural perspective (Fig. 5c). Major structural aberrations are non-existent, as expected for such similar primary sequences. The LRR region does contain numerous site substitutions, some of which are biochemically distinct. Site substitutions are known to have loss-of-function effects in NBS-LRR and could explain the resistance response of MR1[18]. Indeed, the regions ranging from 836-841aa contains multiple, chemically distinct residues. An alternative explanation to protein functional variation is novel or enhanced regulation. (Although both explanations are mutually inclusive.) A 6.1 kb insertion resides 1.5 kb upstream of MR1's NBS-LRR start codon. Such large structural events in enhancer regions are known to be a major driver of phenotypic variation[19].

We confirmed that the community reference genome strain, DHL92, is also resistant (Supplementary Fig. 15). Based on genomic sequences described in this study, the DHL92 serves as a natural recombinant between the *fom2* ORF of MR1 and the insertion-null promoter of AY (Fig. 5). We confirmed these haplotypes with PCR across MR1, AY, and DHL92 (Supplementary Fig. 16). Therefore, it is very likely that the causal variation does reside in the N-terminal tail of the NBS-LRR protein, given that it is substantially altered from AY (Supplementary Fig. 17).

Concerned that FSFHap tends to overpredict parental homogeneity, we also used our more conservative in-house windowing imputer (described above) to examine all variation across this interval. Variant-level inspection of the region reveals how SR-MR1 and PanPipes will have divergent signals around SVs like the one observed in the NBS-LRR promoter (Fig. 5d). AY (susceptible) reads are unable to properly align to this region, resulting in a decline in AY calls. If this insertion was in fact the causal variant, peak association would be

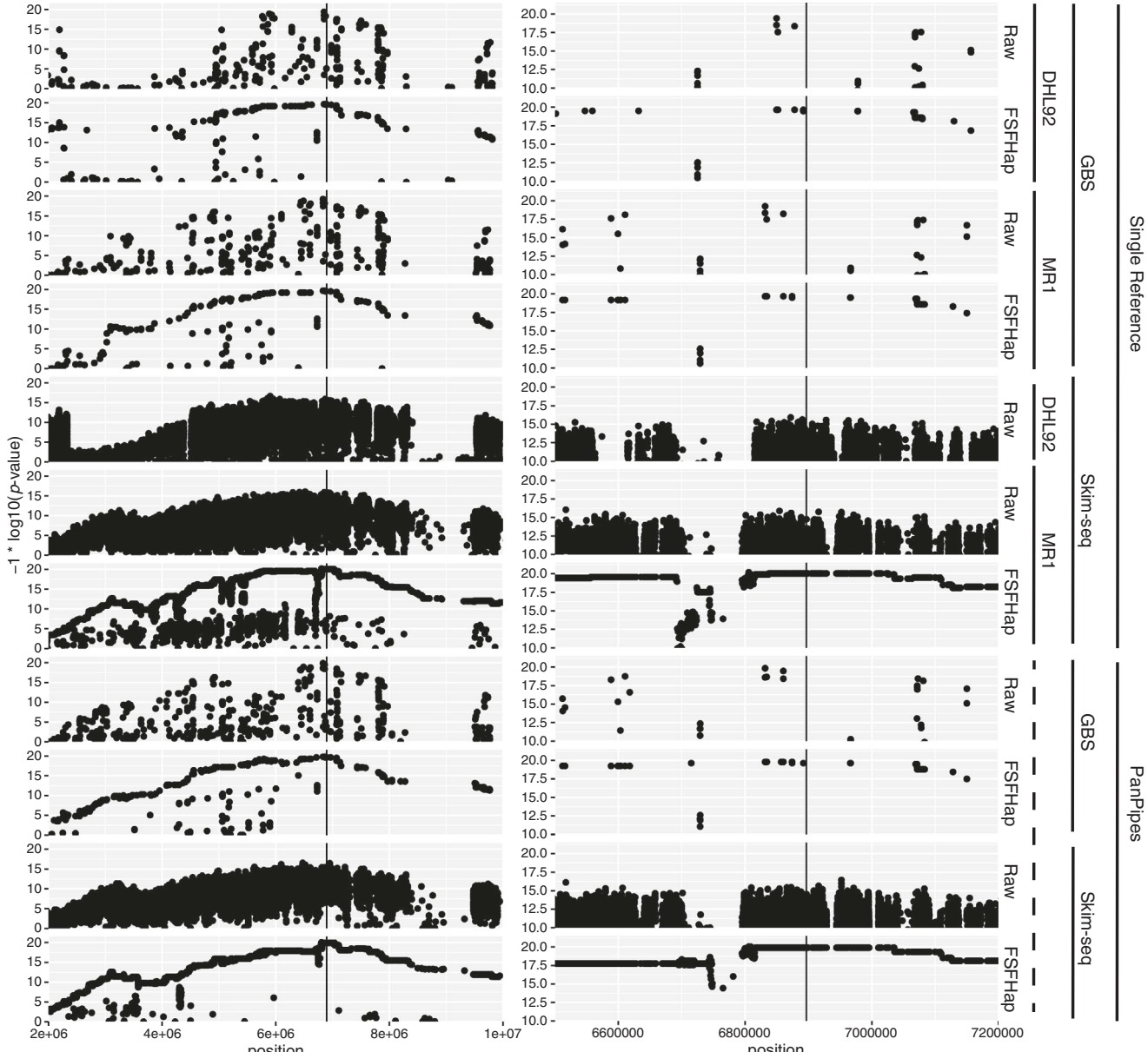

**Fig. 4 | Genetic resolution of the *fom2* locus using assorted methods.** Both panels show association statistic for all variants across the *fom2* interval (see "Methods"). The chemical and analytical genotyping factor combinations are indicated at the far right. Because PanPipes uses both references simultaneously, this factor is shown as a dotted line. "Raw" indicates initial set of filtered variants and FSFHap is the result of imputing the Raw variants. The left panel presents an expanded view of the interval, while the right shows the 700 kb bounding the previously fine-mapped NBS-LRR protein, shown as a vertical black bar in all plots. The combination skim-seq/SR-DHL92 could not be properly imputed and so only "Raw" is shown.

mis-located and its effect underestimated in a low-LD population using an SR-MR1-like approach.

### Gene duplication is the probable mode of Fusarium wilt race-2 resistance acquisition

Resistance to Fusarium race-2 wilt has been previously fine-mapped to Chr 11[13]. In fact, a bacterial artificial chromosome (BAC) of this region specifically from MR1 was previously published as part of that study. As with *fom2* above, the locus contains an NBS-LRR thought to mediate resistance. Interestingly, susceptible lines were also found to possess this NBS-LRR and contain few obvious causal substitutions. In our population, peak association co-localizes with the same NBS-LRR. Detailed examination of the locus suggested that genotype calling in this region was more problematic than the *fom2* locus above. Indeed, there appears to be at least one non-parental

haplotype segregating in two samples. With this in mind, we also explored the window imputed genotypes, which retains all variants but does not impute non-obvious haplotypes (see above). The pattern for association in all cases is centered on a very large 30 kb insertion in MR1 relative to AY (Fig. 6a). The insertion exists in an otherwise highly collinear region of genic space. Dotplots between MR1 and AY indicate this region is the product of an extensive tandem duplication event that has created a second copy of the previously implicated NBS-LRR (Fig. 6b) as well as duplicating four additional viral resistance genes. Protein alignments of the NB and LRR regions of the *fom1*A orthologs and the *fom1*B paralog indicated that the LRR domain has diverged more than any other region in this highly conserved protein (Fig. 6c). LRR domains are thought to be critical in pathogen recognition. We hypothesize that this extensive divergence, in conjunction with natural selection, may have resulted

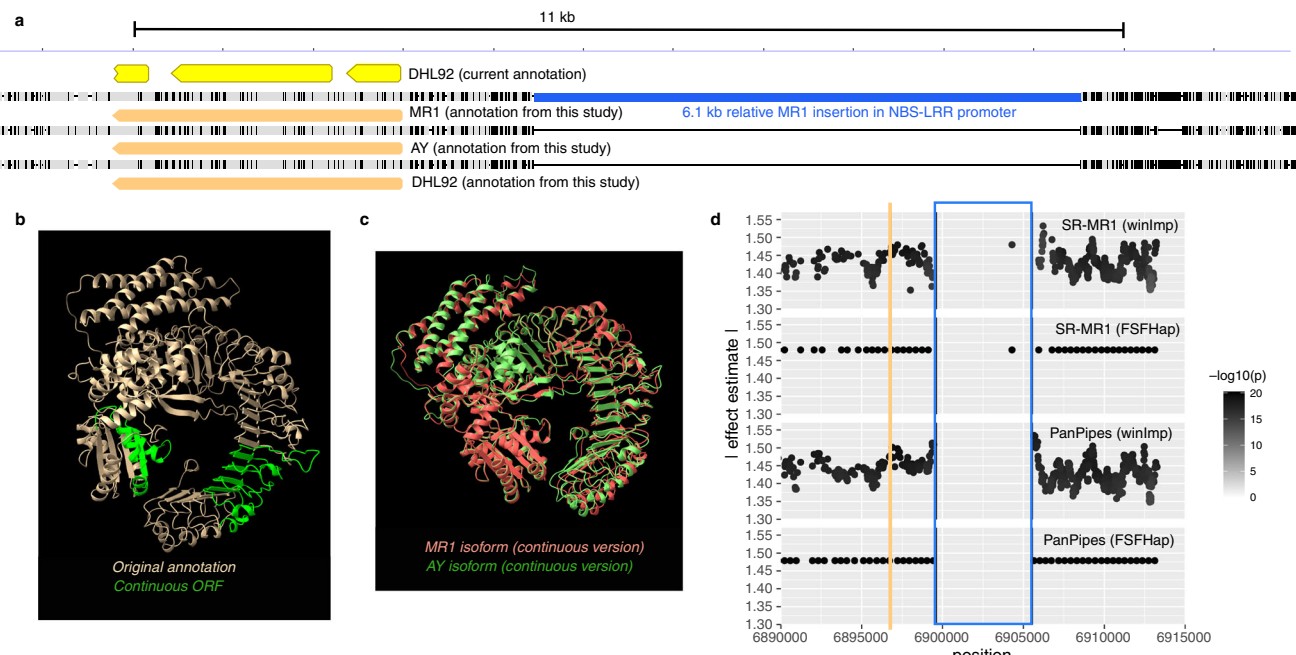

**Fig. 5 | Causal variant analysis of *fom2* locus. a** Nucleotide-level depiction of parent and DHL92 alignments across the *fom2* locus. The pre-existing 3 exon model of the NBS-LRR protein is shown in yellow; our single-exon annotations are shown in orange. The -6.5 kb insertion in MR1's promoter region of the NBS-LRR is shown in blue. Gray indicates identical bases across all three lines. Black lines reflect variant columns. **b** AlphaFold2 predictions of continuous open reading frame (ORF) models with green reflecting the included introns. **c** MR1 and AY isoforms of the continuous model transposed as closely as possible. **d** Allelic effect estimate for variants in the *fom2* region with FSFHap imputation (also depicted in Fig. 4) as well as explicit window-based imputation (winImp) for both PanPipes and SR-MR1 (see "Methods"). Orange bar indicates the center of the NBS-LRR ORF and blue box shows the MR1 insertion span. Variation in winImp calls, relative to FSFHap, results from less aggressive calling algorithm. Source data are provided as a Source Data file.

from error prone repair often observed at tandem duplication boundaries (Fig. 6a).

The originally published BAC does not contain the tandem duplication observed in this study (Supplementary Fig. 18). Manual curation of aligned PacBio reads shows uniform coverage of our MR1 assembly and no aberrant or consistent splits at critical junctures of the repeated sequence (Supplementary Fig. 19). Paralogous sequence divergence is enough to expect >10 kb high fidelity reads to span unique anchor points thereby inferring proper configuration. In contrast, the original BAC was sequenced with 454 technology, which produced ~500 bp reads. Indeed, this BAC sequence is only 70 kb in length compared with larger 172 kb BAC sequenced from a susceptible strain. Although true BAC lengths can vary substantially, it appears more likely that the assembler used in that study choose a path through the conserved region of *fom1* that excludes the duplication entirely. An alternative, though remote, possibility is that the insertion is simply polymorphic in the MR1 strain, and both sequences are correct.

**Powdery mildew resistance loci frequently exhibit recombination suppression and include NBS-LRR hotspots**

The MR1xAY population segregates for three powdery mildew resistance genes (Fig. 7a). Marginally significant peaks on Chr 07 and 11 are related to mild pan-chromosomal LD with these three regions and disappear when multi-locus mixed models are used. The strongest association on Chr 05 presents a mesa-like peak when examined in detail (Fig. 7b). Both imputation methods exhibit comparable profiles, but FSFHap suggests that a small number of recombinants at the 5′ end of the peak are more tightly linked with the trait. The lack of variants in the center of the profile indicates large structural differences between the parents may be suppressing recombination in this region. We examined orthologous regions as a dotplot of both parents. In addition, we overlaid NBS-LRR positional annotations for both genomes (Fig. 7c). Of note, across all three chromosomes harboring strong quantitative trait loci (QTL), only this region of Chr 05 contained evident NBS-LRRs, excepting a single annotation at the 3′ end of Chr 04.

Interestingly, in the case of Chr 05 QTL, the susceptible parent, AY, has retained more NBS-LRR copies than MR1 (Fig. 7c). Most copies are quite divergent, although one NBS-LRR near the largest indel is tandemly duplicated in AY. Two other NBS-LRR clusters within this super-cluster indicate more ancestral tandem duplications/triplications, and these are conserved across AY and MR1. The first cluster, which overlays the highest-effect portion of the peak, has one major SV. This appears to have affected an NBS-LRR exon structure in the first annotation of that cluster, but it is unclear if this is a viable gene in either parent. Other NBS-LRR genes in this cluster are highly similar although rare, biochemically significant substitutions do exist.

Other strong associations with powdery mildew resistance did not contain NBS-LRRs, although Chr 12 QTL has strong precedent in the literature and is proposed to be the gene MELO3C002434, an ankyrin repeat-containing protein[20]. The most strongly associated recombination bin within this interval overlaps MELO3C002434 exactly. Still, in the coding sequence of this gene, there are only 4 nucleotide substitutions between MR1 and AY, and none of these has a substantial impact on the protein product.

The remaining association, on Chr 04, is particularly interesting in that it reflects a very large region of recombination suppression. There are no obvious chromosome-level differences that would drive suppression; in fact, a large portion of this region is far more similar between AY and MR1 than most of the genome. Regardless, such patterns have clear implications for further fine-mapping or introgression of this region into other germplasm.

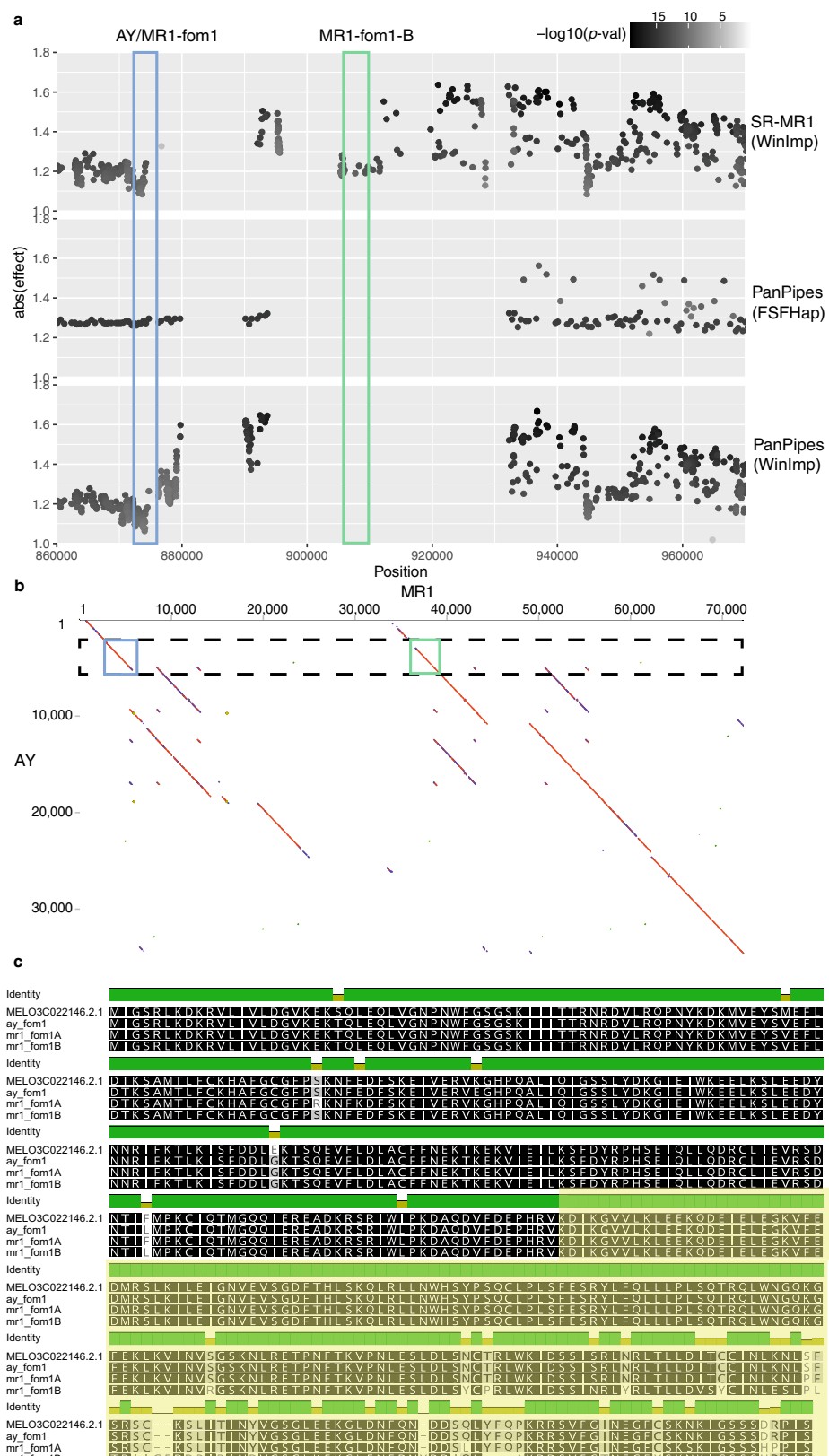

**Fig. 6 | Substantial tandem duplication of the *fom1* locus on Chr 09.**
**a** Association profiles across various genotyping/imputation methods described in main text. Estimated effects are depicted on *y*-axis and darkness of dot indicates *p* value (see "Methods"). Blue box highlights the known *fom1* NBS-LRR while green box defines paralog position identified from this study. **b** Dotplot between AY and MR1 (origin at top left) shows substantial duplication of large NBS-LRR containing

region. Colored boxes are consistent across **a** and **b**. **c** Amino-acid alignment of DHL92 annotated NBS-LRR, AY Fom1 translation, and both paralogs of Fom1 translation from MR1. The LRR region is highlighted in yellow and the NBS domain is unshaded. The N-terminal TIR domain is not shown due to space constraints. Variation in winImp calls, relative to FSFHap, results from less aggressive calling algorithm. Source data are provided as a Source Data file.

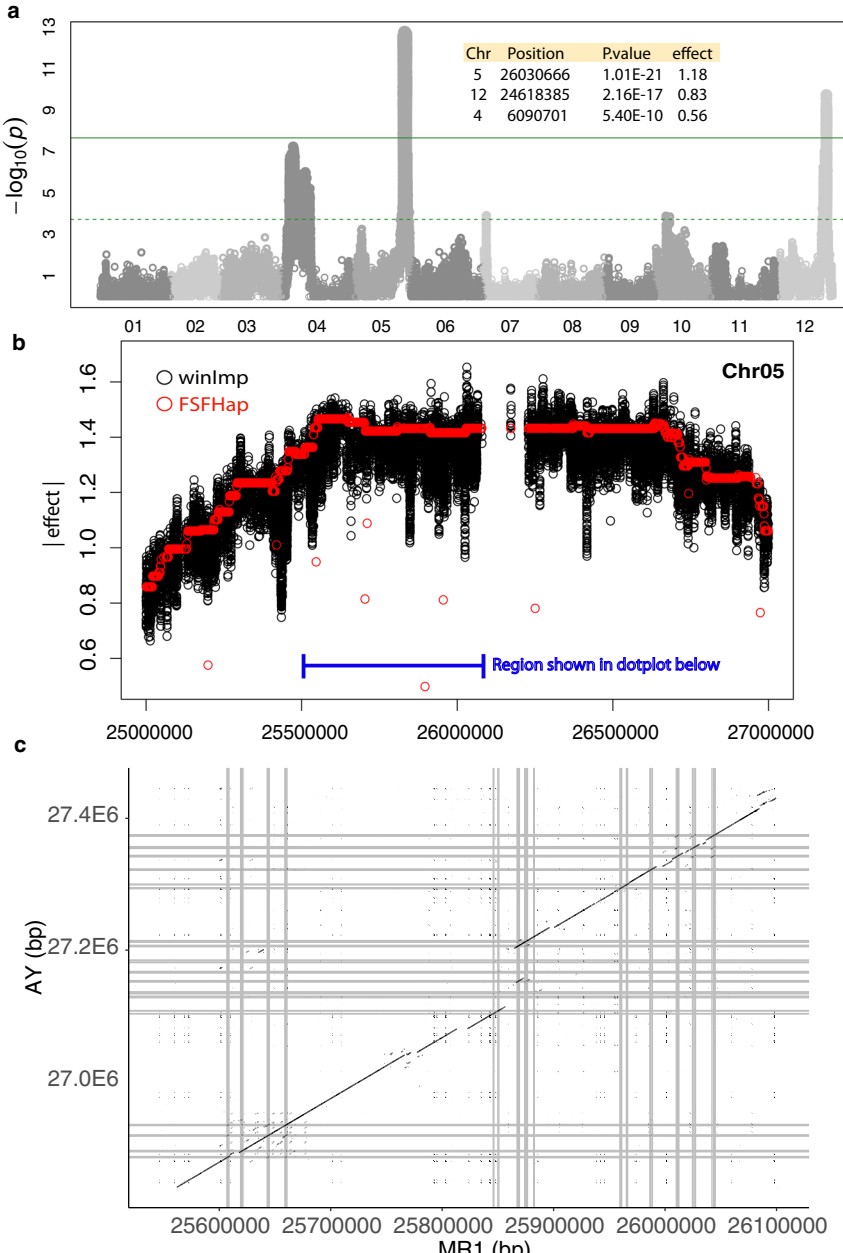

**Fig. 7 | Rampant NBS-LRR turnover underlying powdery mildew resistance.**
**a** Genome-wide association profile for powdery mildew resistance. The dashed and solid line indicated the false discovery rate and Bonferroni thresholds, as calculated by GAPIT3 (see "Methods"). The green solid line indicated the Bonferroni < cutoff. Inset table describes salient features of lead variants based on winImp genotypes.

**b** Detailed view of estimated effect size around the Chr05, large-effect QTL under both winImp and FSFHap imputation. **c** Dotplot of association plateau region between AY and MR1. Gray bars indicate positions of NBS-LRR homology. Source data are provided as a Source Data file.

## Discussion

Low-error, long-read sequencing has given individual researchers the capacity to create their own reference genomes for use in genetic analysis. In this study, we examined what experimental gains might be achieved even in the simplest type of genetic experiment: an advanced generation RIL population formed from a biparental cross. Any gains observed in such a simple case should be amplified in more complex experiments, such as MAGIC crosses and unstructured diversity panels. Recent work in a complex tomato population support this assumption, where the use of a variation graph enabled the assignment of 25% more phenotypic variation to genetic loci than single-reference approaches[21]. The simple pedigree in our study allowed us to further characterize genotyping efficacy under various molecular and bioinformatic scenarios.

Many communities studying non-model organisms, such as cucurbit species, continue to use reference sequences produced primarily from short-read technologies. Numerous observations in this study reveal the inevitable errors and limitations of such sequences. The original short-read assembly of the melon reference strain DHL92 is clearly useful for low-resolution mapping but its divergence from parents and incompleteness can blur the full genetic resolution of an experiment, even with the modestly scaled population used in this study (Fig. 4). Under fine-grained analysis, the capacity to accurately assess large SVs, particularly tandem duplications, requires long-read data (Fig. 1). A long-read assembly of DHL92 was recently reported, supporting this result as well[15].

Beyond improvements in pre-existing references, the capacity to compare multiple references is useful in interpreting genetic

associations. SVs will certainly not always be causal mutations underlying a phenotype, but they do appear to be much more likely to cause both major and minor effects[7,19,22]. The parent sequences used here allowed us, in effect, to completely impute high-quality genome sequences of all individuals in the population. Our *fom1* results illustrate the significant gains achieved with a comparative genomics perspective on association results: if only MR1 or only AY had been sequenced, the *fom1* region would have been very challenging to interpret (Fig. 6a). Indeed, in the AY-only case, most markers would have been filtered out due to excessive heterozygous calls commonly triggered by tandem duplications. While graph-based approaches, such a PanPipes, have the capacity to encode such information as cyclic loops in a graph, we found such loops very difficult to interpret during the genotyping phase. To that end, PanPipes treats such duplications as indels. As such, after-the-fact dotplot analysis is clearly vital to base-level interpretation (Figs. 1c, 6 and 7).

Graph-based techniques have been shown here and elsewhere to improve proper read placement, but they represent a substantial break from the standard bioinformatic ecosystem. Many of the formats and methods used are rapidly changing and lack downstream analytical and visualization support. We explored possible benefits in gene and causal variant discovery relative to these costs. As expected, all methods were able to discover single, large-effect loci (Fig. 4). Although PanPipes offered the clearest interpretation of these peaks, SR-MR1 approach was comparable. PanPipe's major advantage is in the sheer number of variants it can discover *and* genotype (Fig. 3). Manual curation around SVs suggests that these variants and, more importantly, their surroundings are hotspots for differential performance, as expected of graph-enabled alignment[1]. Indeed, in a GWAS context where an SV is the causal variant, the two methods can produce opposing patterns of association if imputation fails. In such cases, reference bias will skew apparent allele frequencies and weaken significance relative to adjacent variants. In experiments where imputation is not possible at all, such as QTL-seq, these issues will be inevitable and graph-base approaches will be more accurate.

Whereas bwa-mem aligns to quite divergent sequence, giraffe has a much higher stringency because most variants are expected to be in the reference graph. In fact, the stringency goes beyond variant specific alignment: giraffe makes graph-alignment feasible by relying on haplotype information when areas of the dense variants would result in combinational explosion. Manual inspection of evident SR-MR1 genotyping errors often indicated that the read triggering the call had an MQ = 60 and a full-length alignment. Such calls are in fact sequencing error that are coincidental on the incorrect allele. Given 1x coverage and an error rate between 1:100 and 1:1000, this will equate to ~6 miscalls per 10,000 calls. In some cases, giraffe penalizes such reads because they fail to phase with the haplotype and, thus, such reads are thrown out. Though this does not completely account for the difference in error rate, it could comprise a sizeable fraction. Many such errors will be flipped to the proper base during imputation and thus, for operational purposes, the SR-MR1 error rate will be closer to PanPipes. Yet, the imputation accuracy itself will inevitably benefit from higher stringency as haplotypes become more diverse and LD blocks become narrower in more complex populations.

We have chosen a fully graph-based pipeline to contrast with conventional SR approaches (Fig. 2), but numerous hybrid approaches could be imagined[2]. For a low-resolution, biparental mapping, standard SR genotyping and association using the community reference could be paired with case-by-case examination of orthologous regions across parental assemblies. In this study, such an approach, particularly if using skim-seq, would have produced insights comparable to the full, graph-based analysis. Still, the dense genotyping afforded by the graph helps to assure that complex variants are legitimate. Otherwise, such confirmation would require more extensive follow-up molecular analysis.

Though models of NBS-LRR biochemistry are being actively refined, many lines of evidence point to the LRR region of these proteins as direct-effector binding sites in fungal resistance pathways[23,24]. Our results related to *fom1* support a sub-functionalization model in which tandem duplication has generated an NBS-LRR paralog (in MR1) that, either via the duplication process or subsequent mutations, underwent radical alteration to achieve resistance to novel strains of *Fusarium*. In contrast, adaptive mutations in the MR1 *fom2* allele did not require duplication but appear to be the result of a suite of mutations in the LRR domain—as with *fom1*. Interestingly, the promoter region (-1.5 kb upstream) of the gene appears to be robust to the introduction of a large insertion in MR1. Since fungal responsive NBS-LRRs are typically constitutive "front-line" sensors, simple enhancer/promoter structures may be robust to such major mutations[25].

Our results also give perspective on the role of in silico structure prediction in variant analysis. AY and MR1 NBS-LRRs were predicted to be effectively identical within the bounds of variation generated from recurrent rounds of prediction (Fig. 5). This similarity occurs despite a striking switch in hydrophobicity in numerous LRR-domain residues (Supplementary Fig. 17). It remains unclear if these in silico structures are legitimate, or if distinct behavior is solely a product of side-chain reactivity.

We observed extreme enrichment in NBS-LRR genes in the largest effect association with powdery mildew resistance. This region exhibits a low recombination rate, which may be a product of the structural divergence between these two parents (Fig. 7). Interestingly, the quantity of NBS-LRRs in this region is not correlated with resistance, since AY has ~30% more than MR1. The strongest association lies in the first cluster of NBS-LRRs, in which both parents have 4 genes but MR1 has a 7 kb insertion.

To conclude, the availability of all variation in this population has proven extremely useful in interpreting causal loci, particularly structurally divergent disease loci. This comparative-genomics perspective does not necessarily require the implementation of graph-based methods, but the enrichment of informative variants benefits even low-resolution biparental mapping. Such benefits will be amplified in populations with far more complex patterns of LD. However, the full implementation of graph-based methods remains a bioinformatic challenge. As part of this manuscript, we propose our PanPipes strategy, focused on a skim-seq approach (Fig. 2). We have released all relevant software free to the public[26]. Effort has also been made by other groups toward visualization and annotation of variation graphs[27]. While these tools are centered on human genetics and remain challenging to implement, they encourage further development in agricultural and non-model contexts.

## Methods

### Plant materials

MR-1 (MR1 hereafter), a multi-disease resistant melon (*Cucumis melo* subsp. *melo*) line was derived from an Indian landrace melon, USDA PI 124111[28]. Ananas Yoqne'am (AY) is an heirloom Israeli melon cultivar (*Cucumis melo* subsp. *melo*) that is highly susceptible to numerous plant pathogens[29]. A RIL population ($N = 149$) was developed by making an initial cross of a female MR-1 by a male AY. The resulting $F_1$ was self-pollinated. Individual $F_2$ plants were carried to the RIL stage ($F_6$–$F_{11}$) through single seed descent by manual self-pollination. A total of 149 RILs were used in this study.

### Sequencing and assembly of MR1 and AY

High molecular weight gDNA was isolated through fee-for-service with Polar Genomics LLC (Ithaca, NY). The gDNA was sheared with a gTube to an average fragment length of 13 kb. Sheared gDNAs were converted to a library with the SMRTBell Express Template Prep kit 2.0. The library was sequenced on 1 SMRTcell 8 M on a PacBio Sequel II using the circular consensus sequencing mode and a 30 h movie time. CCS

analysis was done using SMRTLink V8.0 with default parameters. Reads were assembled using HiFiASM (v0.5) using inbred flag (-l0)[30]. Resultant contigs were initially scaffolded using RagTag (v1.0.1) (https://github.com/malonge/RagTag) and the community reference, DHL92_v3.6.1[31]. An additional round of scaffolding was performed using RagTag and the other parent: AY using MR1 and MR1 using AY. Assembly results (Table 1) were also cross-checked with the Canu assembler (v2.0)[32] and final statistics determined with BBTools (v38.87)[33] and BUSCO(v4) with embryrophyta_obd10 dataset. SVs associated with *fom2* were confirmed via PCR using standard protocols (Supplementary Data 2).

## Gene annotation

For each parental genome separately, k-mer counts were extracted from Illumina 150 bp reads (see below) using Jellyfish (v2.2.10)[34]. The k-mer counts were provided to GenomeScope v2.0 to determine the haploid "kcov" value[35], which was doubled to represent the diploid kcov value. Repetitive k-mers were masked using Kmasker v1.1.1 rc231015 with a repeat frequency threshold of 5x the diploid kcov value[36]. A soft-masked genome was produced using bedtools v2.27.1 and the Kmasker repeat gff. The soft-masked genomes were annotated using Braker v2.1.6 separately with species proteins and RNA-Seq reads[37]. The species protein database was constructed from cucurbit peptides (ftp://cucurbitgenomics.org/pub/cucurbit/genome/). The RNA-seq reads used are available at NCBI, BioProject accessions PRJNA358655 (MR1) and PRJNA358674 (Top Mark/AY). The RNA-seq reads were mapped using HISAT2 v2.2.1. The BRAKER annotations obtained using proteins and RNA-Seq were combined using TSEBRA v1.0.2[38]. Additional k-mer-masked annotation details are available at Github (https://github.com/brianabernathy/kmer_masked_annotation).

## Pangenome graph generation with MR1, AY, and DHL92

Dotplots of resulting pseudomolecule scaffolds (Supplementary Figs. 1–12) were examined to validate that all chromosomes were col-linear and did not contain inter-chromosomal translocations, although intra-chromosomal translocations were permitted. All homologous pseudomolecules from each of the three genomes were aligned in a chromosome-wise fashion using progressiveMauve algorithm (2015-02-13 linux release)[39]. Default parameters were used except seed weight was altered to 27 based on manual evaluation of parameter sweeps across seed weight (-seed-weight) 17 to 31 and weight (-weight) default to 10,000. progressiveMauve infrequently produces false gap openings that are easily identified by adjacent gaps of equal size. All such gaps were removed using xmfa_tools, published as part of the PanPipes suite (see "Code availability" section). Chromosome align-ments were converted to a graph format (GFA) using xmfa_tools and vgtools. These graphs were used in conjunction with vg giraffe for all downstream read mapping and genotyping[5].

## Skim-sequencing MR1xAY population

The shotgun genomic libraries for each individual in the population were prepared with the Nextera Flex sample prep kit from Illumina. The libraries were pooled, quantitated by qPCR and sequenced on two SP lanes for 151 cycles from both ends of the fragments on a NovaSeq 6000. Fastq files were generated and demultiplexed with the bcl2fastq v2.20 Conversion Software (Illumina). bbtools (v38.87) was used to assess quality and trim adapter reads.

## Graph-based genotyping

Both skim-seq reads and GBS reads[12] for the RIL population were aligned to the graph generated above using giraffe[1], available in vgtools (v1.37.0), following the pipeline based on index construction from a GFA file with embedded path information, which were in this case MR1, AY, and DHL92. Genotyping was performed by first identi-fying all bifurcating branchpoints in the graph, read in the forward direction. Hyper-variable regions were defined as 100 bp windows (relative to linear MR1 sequence) with >15 variants; these regions were masked in subsequent analysis. Only reads with Mapping Quality (MQ) = 60 were retained. Alleles were then called based on the read coverage of the two alternative edges—in effect, two possible alleles—as assessed by vg pack's resultant edge table. Calls were made based solely on the allele with the most coverage, since, for RILs, hetero-zygosity should be low and would be evaluated more effectively during imputation.

## Single-reference genotyping

Trimmed reads described above were aligned to reference genomes using bwa-mem (bwa v0.7.17-r1188) with default parameters[40]. Resul-tant BAM files were jointly supplied to mpileup (bcftools v1.14)[41] and allele calls were reduced to haploid calls using allelic depth field and majority rule, as above, to coincide with PanPipes methodology as closely as possible.

## Imputation

Imputation was done in two ways: (1) FSFHap[17] (TASSEL v5.0)[42] was used directly on GBS data and skim-seq data down-sampled to less than 1 variants per 700 bp and all GBS calls. (2) For the full (not down-sampled) skim-seq dataset, a custom algorithm (see "Code availability" section) was implemented such that, for a window size of 31 variants, the central allele was called as its consensus haplotype if 92% of the haplotype matched one parent and 30% of the alleles were explicitly called. Central alleles in cross-over haplotypes were called by majority rule. Otherwise, a haplotype was considered un-imputable. If less than 4% of samples had un-imputable flags for a region, un-imputable samples were called heterozygous. Otherwise, the affected sample was left unimputed.

## Comparison with single-reference MR1 and DHL92 approaches

It was evident from initial analysis of GBS data that lines RIL194 and RIL37 had highly aberrant genotyping patterns and were therefore excluded from genotyping comparison. For the remaining samples, genotyping accuracy was assessed using the parental assignments for each individual made during imputation. The explicitly called allele was compared with the imputed allele. Only variants that had differing, explicit calls between parents were used and then only if the call for the individual sample was also explicitly defined, i.e., no variants imputed to be heterozygous or poorly called parental regions were included. The percentage of mismatch was reported relative the total compar-isons passing these criteria.

## Genetic association

Phenotypic data for *Fusarium* (race 1) and powdery mildew resistance was drawn from previously described work[12,43]. *Fusarium* (race 2) experiments were performed according to those protocols[12]. In brief, propagation trays were filled with trimix that was saturated in spore suspensions. Unsaturated control trays were also prepared. Five seeds per RIL per tray well were replicated twice. Twenty-eight days post-inoculation, RILs were scored from 1 to 5 on severity, where 1 was asymptomatic and 5 was dead. For association, we used post-imputed genotypes to test each variant's estimated effect using a generalized linear model with additive variance since few if any heterozygotes were present to estimate dominance. To control for possible cryptic/ biased relatedness or multi-locus effects, we also confirmed resultant peaks using both a standard mixed-effects model as well as FarmCPU[44]. All algorithms were implemented in R (v3.4.1) using GAPIT3 (v2020.10.24)[45] with default parameters. This approach allowed us to include lines RIL194 and RIL37 for mapping. Both lines did have informative parental segregation for at least half of the genome; the remainder of sites were either treated as heterozygous or left unim-puted. Association profiles are based on raw *p*-values or absolute

effects using the generalized linear model. Unless thresholds are shown (e.g., Fig. 7), all points in association profiles exceed both false discovery rate and Bonferroni thresholds as calculated by GAPIT3.

### In silico protein prediction
The protein structure of eight sequences was predicted computationally using Alphafold v2.0.0[46]. The four test sequences for *fom2* were: (1) the spliced isoform from DHL92, (2) the continuous isoform from DHL92, (3) the continuous isoform from MR1, and (4) the continuous isoform from AY. The four negative control sequences were also included as shuffled or reversed proteins in both DHL92 continuous and spliced versions. The databases used for structure construction were the Uniref 90 database from August 2018 (https://ftp.uniprot.org/pub/databases/uniprot/previous_major_releases/release-2018_08/), the Mgnify database from December 2018 (ftp://ftp.ebi.ac.uk/pub/databases/metagenomics/peptide_database/2018_12/mgy_clusters.fa.gz), the Uniclust 30 database from August 2018 (https://wwwuser.gwdg.de/~compbiol/uniclust/2018_08/), and the BFD database "bfd_metaclust_id30_c90" from March 2019 (https://bfd.mmseqs.com/). The preset search setting "full_dbs" was used for homology search and folding.

### Reporting summary
Further information on research design is available in the Nature Portfolio Reporting Summary linked to this article.

## Data availability
Raw data and genome assemblies have been deposited to NCBI under Bioproject accession PRJNA844271. Source data are provided with this paper.

## Code availability
Computational tools developed in-house to support this analysis have been published under a creative commons license at Github (https://github.com/USDA-ARS-GBRU/PanPipes)[26].

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

## Acknowledgements

We thank Brian Scheffler, Jeremy Edwards, and Grant Billings for preliminary review and suggestions. We also thank Adam Novik and Jouni Sirén for assistance on vgtool usage related to unique aspects of the PanPipes framework. This research was funded by Agricultural Research Service USDA projects: 6066-21310-005-00-D (J.N.V., A.M.H.-K., A.R.R.), 6080-21000-019-011 (S.E.B., W.P.W., A.L.), 6080-22000-028-00-D (S.E.B., W.P.W.).

## Author contributions

J.N.V. wrote manuscript, developed software, designed project, and analyzed data. S.E.B. designed project, wrote manuscript, and performed experiments. B.A. wrote software and analyzed data. A.M.H.-K. generated and analyzed data. A.R.R. generated and analyzed data. A.L. performed experiments. W.P.W. designed project, wrote manuscript, and performed experiments.

## Competing interests

The authors declare no competing interests.
