## [Peer Review File · Nature Communications]

Graph-based pangenomics in a non-model crop maximizes genotyping density and reveals structural impacts on fungal resistanceReviewers' Comments:

Reviewer #1:

Remarks to the Author:

This manuscript describes innovative work to develop and utilize graph-based pangenomics approaches for melon (*Cucumis melo*) and demonstrates value of those approaches to identify and resolve genetic variants responsible for important disease resistance traits. Long-read (PacBio) sequence was used for de-novo assembly of parental genomes, which together with the reference melon genome, and skim sequencing of a biparental RIL population, was used to establish a pan genome. Skim sequencing of the biparental RIL population also was used for QTL analysis combined with pan-genome based fine genetic analysis to identify likely causal mutations and distinguish between alternate possibilities.

Below are some places where additional information or clarity would be helpful.

1. "Our pangenomic pipeline is robust to large-scale chromosomal variation if major inter-chromosomal translocations are not present, so we did not interrogate these inversion/translocations further". what is the basis for making this claim (that is is robust to large-scale chromosomal variation)? Are we to conclude that there were not inter-chromosomal translocations? Why were the inversion/translocations in this region not interrogated? Were they not of concern relative to the specific traits of interest in this study? If so, would be helpful to say so.
2. The use of variants from the segregating population to filter variants present in the PanPipes assembly is interesting and appears to be useful (Fig3), but it was unclear why a similar result would not be obtained from the original comparison of the two parental genomes (AY, MRI) relative to reference genome (DHL92)? Was the low coverage referred to in the sentence "This difference makes initial variant numbers for SR-MR1 and SR-DHL92 more difficult to evaluate because low-coverage data triggers numerous false variants from sequencing errors" referring to the PacBio reads? (if so, please state)
3. Fig 4 right panel shows a distinct gap at ~6.75 Mb. Please comment on this.
4. Fig 5D. 'if the insertion was the causal variant, peak association would be mis-located'. Where would it be mis-located to? From panpipes Winimp panel it seems that the peak is occurring on either side of the insertion.
5. Fig 6A. "the pattern for association in all cases is centered on very large ...insertion in MR1". this was unclear as the insertion was not evident from the PanPipes panels. please clarify. What does this say about relative usefulness of the different methods?
6. Fig 7B. 'the lack of variants in the center of the profile indicates large structural differences...'. Were large structural differences found? (since this is also mentioned in the discussion, it would be helpful to elaborate on it)
7. Fig 7C. Please tell the reader in the text that Fig 7C is looking at the plateau region (and not the gap discussed in the prior sentence). What should the reader be seeing from Fig 7C? If I understand fig 7B correctly, the 'recombinants at the 5' end of the peak that are more tightly linked' are located at 25.5-25.7 Mb but 7C is considerably broader, 25.6-26.1. It would help the reader to annotate the relevant regions in both 7A and 7C so they know where to look, and to tell the reader why the broader region is shown in fig 7C.
8. Discussion section (powdery mildew). "The strongest association lies in the first cluster of NBS-LRRs, in which both parents have 4 genes but MR1 has a 7kb insertion." This insertion was not mentioned in the results section, please mention and indicate where it is.

Other items:

1. Table 1. Are the assembly data in Table 1 from PacBio sequencing (and only PacBio)? Please clarify in the table title.
2. Fig 1B and 1C are not mentioned in the text. They should be described and discussed.
3. "We confirmed that the community reference genome strain, DHL92, is also resistant". Please inform the reader in this sentence that DHL92 does not have the 6.1 kb insertion.
5. Fig 7C. it would be helpful to tell the reader the number of NBS-LRR copies in each parent in the

results section.

Reviewer #2:

Remarks to the Author:

The manuscript describes use of pangenome graph for analysis of bi-parental population and QTL discovery in melon. It also introduces a new pangenome construction and analysis tool/pipeline – PanPipes.

I would like to start with the aspects of the manuscript I really enjoyed

- The manuscript is topical, the concept of pangenome graphs and their use as a reference is making its way into the plant genomic community
- The authors centre their analysis on application of pangenome graphs for analysis of bi-parental population sequenced using skim-Seq/GBS and the advantage of this approach over using a single community reference. Bi-parental populations are commonly used for QTL analysis in crop plants. The authors provide a clear, tangible example of a strong advantage of using pangenome graph and how it can practically improve genomic analyses
- The manuscript includes an in-depth discussion of three disease resistance/susceptibility loci. Each represent a different scenario of acquiring resistance/tolerance.
- The manuscript is very clearly written, making the explanations accessible even to the non-expert reader

My main major concern that it is hard to imagine that the PanPipes pipeline will be widely used by the community because it relies on a very old aligner, whose support has long been discontinued (<https://darlinglab.org/mauve/mauve.html>). There is no way of knowing how Mauve will perform on larger more repetitive genomes common in crop plants (soybean, wheat, barley; at least I don't know of any benchmarks which included it). On the GitHub page authors state 'Software, such as pggp, is a recent method to generate such graphs. In our hands, we have noticed that pggp often generates unwarranted cycles that violate the positional homology paradigm followed in PanPipes (see above). Our current preferred method is to rely on a multi-threaded implementation of the progressiveMauve algorithm, see GPA.' Which sadly makes the PanPipes already outdated.

I think PanPipes should be updated to at least allow passing of results from modern aligners. xmfa_tools allow conversion of Mauve xmfa file format into gfa, but probably the most common sequence alignment formats these days are maf/paf (I am including paf format here as in principle for bi-parental populations graph can be built with just two genomes and minimap2 and wfmash produce paf). There should at least be utility to convert maf and paf to xmfa and check if the gfa files (built by xmfa_tools using other aligners or other pipelines [pggp, cactus pangenome pipeline, minigraph]) meet the requirements to be used in PanPipes (and if they don't what needs to be done to get there).

Other, smaller concerns:

We hypothesize that this extensive divergence, in conjunction with natural selection, may have resulted from error prone repair often observed at tandem duplication boundaries.

This statement needs a reference.

Reviewer #3:

Remarks to the Author:

The authors sequenced and assembled two lines of melons, MR-1 (multi-disease resistant) and AY (susceptible), using long reads. The authors genotyped a RIL population of N=149 and did the genome-wide association analysis of the disease resistance using both the conventional single-reference approach and the pangenome approach. During the analysis, they found several

bioinformatic issues related to (I believe) the lack of mature ecosystem of pangenome analysis. To address (at least) some of the issues they found, the authors developed a new pipeline, PanPipes, that works with shallow coverage sequencing data better than existing methods. The authors analysis on disease resistance revealed that (putative) causal structural variations are more easily found with PanPipes and the pangenome approach.

The results suggest that the pangenome approach is a more powerful tool for agricultural and other non-model species.

It was a pleasure to read this paper. The paper brought me to a journey in which I see many bioinformatic issues the authors found during the analysis, which will shed light on challenges we, the genomics community, have to address in the near future. Not only the authors propose solutions to some of these issues, but also, they spur future development of pangenome analysis tools. More specifically, the graph-genome aligner challenge (ignoring too repetitive seeds, ignoring hypervariable regions for avoiding combinatorial explosion), the DAG challenge (tandem duplication has to be represented as an insertion because loops cannot be handled by the current tools, etc.), and the gene annotation challenge (how to annotate genes more accurately without prior knowledge of repetitive elements in the genome). I am not sure if the authors want to sell these, but however, I believe that this paper is going to be a landmark paper that will be cited dozens of times (or maybe hundreds) by the tool developer community.

Put aside pointing out these challenges, they authors presented a model case where PanPipes finds a causal region for binary trait, using biparental cross, which itself is intriguing. The conclusions are well supported by experimental evidence where possible.

Here are some minor issues:

1. Gene annotation section in Materials and Methods seems to lack a citation to TSEBRA (Lars Gabriel, et al, BMC Bioinfo. 2021)

2. Graph-based genotyping section says

> Hyper-variable regions were defined as 100 bp windows (relative to linear MR1 sequence ... were masked in subsequent analysis.

Adding a bit of rationale for this procedure would help readers understand why this is necessary, though I guessed that this was for avoiding false positive calls.

3. In Data and software availability section, the authors state that the software is licensed under Creative Commons without specifying a variant, though the Creative Commons license have several variants. Taken together with "We have released all relevant software free to *the academic public*" (emphasis by me) in Discussion, the statement would lead to misunderstanding such as PanPipes is licensed under CC BY-NC-SA, which is a more restrictive license than CC0, which is indicated in the GitHub repository. My suggesting is to specify CC0

4. Fig 1C must indicate a region of Ns in some way (my suggestion is to show it as a gray rectangle in the dotplot). If I understand correctly, the dotplot in the middle must have a region of Ns, which appears like insertions in DHL92, which confused me for a few minutes.

5. The second paragraph of the result section says "We observed that conventional repeat modeler approaches..." though it was unclear to me what the "repeat modeler approaches" are. Is that an approach that we find repeat sequences by RepeatModeler2 (Jullien M Flynn, et al, PNAS, 2020) or EDTA (Shujun Ou, et al, Genome Biol, 2019)?

The following are just my comments, not issues.

The fact that the quantity of NBS-LRRs is not correlated with the disease resistance was intriguing because it underscores the importance of graph genome analysis. Conventional approaches such as presence-absence association analysis would not be able to reveal such an association. Also, Fig 6 & 7

are impressive.

The paper convinced me that every reference genome should be reassembled from long reads. Given the timeframe of this study, it might have been difficult to reanalyze everything using the new reference genome by long-read sequencing [33]. However, this is not asking the authors to reanalyze, because I believe such an analysis would not affect the final conclusion very much.

Response to Reviewers for “Graph-based pangenomics . . .” (NCOMMS-22-28044-T)

Note, all line numbers are based on resubmission DOCX files and changes are highlighted therein. The revised main manuscript PDF does not contain line numbers.

Reviewer #1 (Remarks to the Author):

This manuscript describes innovative work to develop and utilize graph-based pangenomics approaches for melon (*Cucumis melo*) and demonstrates value of those approaches to identify and resolve genetic variants responsible for important disease resistance traits. Long-read (PacBio) sequence was used for de-novo assembly of parental genomes, which together with the reference melon genome, and skim sequencing of a biparental RIL population, was used to establish a pan genome. Skim sequencing of the biparental RIL population also was used for QTL analysis combined with pan-genome based fine genetic analysis to identify likely causal mutations and distinguish between alternate possibilities.

Below are some places where additional information or clarity would be helpful.

1. “Our pangenomic pipeline is robust to large-scale chromosomal variation if major inter-chromosomal translocations are not present, so we did not interrogate these inversion/translocations further”. what is the basis for making this claim (that is is robust to large-scale chromosomal variation)? Are we to conclude that there were not inter-chromosomal translocations? Why were the inversion/translocations in this region not interrogated? Were they not of concern relative to the specific traits of interest in this study? If so, would be helpful to say so.

Sorry for glossing over this point; we have attempted to clarify the manuscript [ln 261-265]. To elaborate, the multiple sequence alignment method used (progressiveMauve) does not force global collinearity but allows the alignment to be broken into large, locally collinear blocks. Therefore, a large inversion will be treated as homologous genomic space. One of nice things about the graph-based approach is that we can recover variants from within these inversions and acknowledge the major event as well.

Though we did not detect any substantial inter-chromosomal events, we wanted to empathize that these would need to be accounted for if they did exist. In theory, this could be done through concatenating the affected chromosomes. We describe this to some extent in the PanPipes github site.

By “not interrogate”, we meant we did not confirm these via PCR, Hi-C, etc. As surmised by the reviewer, had the region shown any associations with the traits under study, we would have investigated it further.

2. The use of variants from the segregating population to filter variants present in the PanPipes assembly is interesting and appears to be useful (Fig3), but it was unclear why a similar result would not be obtained from the original comparison of the two parental genomes (AY, MRI) relative to reference genome (DHL92)? Was the low coverage referred to in the sentence “This difference makes initial variant numbers for SR-MR1 and SR-DHL92 more difficult to evaluate because low-coverage data triggers numerous false variants from sequencing errors” referring to the PacBio reads? (if so, please state)

Yes, we agree that was confusing. This is one of the more difficult (but important) aspects of comparing single reference and graph-based approaches. In the SR approach, variants are called jointly, i.e. all samples are used, such that even a single read in a single RIL will trigger a variant. As you might expect, many of these are junk when using low coverage data because Illumina still has a 1% error rate. Therefore, we felt that the fairest approach was to start at a point of comparison where the “obvious” junk variants had been removed. Because this is a well-structured population, expected segregation pattern was the most powerful initial filter. We have restructured this paragraph [ln 368-375].

3. Fig 4 right panel shows a distinct gap at ~6.75 Mb. Please comment on this.

Interestingly, this region is a highly conserved monomorphic stretch of approximately 20kb. We now make a note in the figure legend.

4. Fig 5D. 'if the insertion was the causal variant, peak association would be mis-located'. Where would it be mis-located to? From panpipes Winimp panel it seems that the peak is occurring on either side of the insertion.

We agree this point was not well articulated. We have substantially revised [ln 526-535]. Namely, we were attempting to make a broader point using these association profiles as an example for those who might be working with a much less structured, less "imputable" population. Since WinImp does not force an explicit model onto data, we recover true signal from any additional haplotypes and also (more likely in this case) genotyping variability, thus reflecting a more GWAS-like situation. We think the relative behavior in this case is quite interesting even if expected from initial results.

5. Fig 6A. "the pattern for association in all cases is centered on very large ...insertion in MR1". this was unclear as the insertion was not evident from the PanPipes panels. please clarify. What does this say about relative usefulness of the different methods?

Here we were referring to the trough that occurs in the SR-MR1 profile around the expected causal gene. In the PanPipes profile that gene is in a large insertion but flanked by clear strong association. Taken together, we feel this makes the PanPipes profile more interpretable since the insertion has similar associations at both ends. We have tried to clarify [ln 548-549], and welcome additional suggestions.

Your point about relative usefulness is particularly interesting in this case of tandem duplication (TD). The genome aligner must choose the "best" path through the similiary matrix, and this dictates the variants called by PanPipes. In the case of a segregating TD recombining with various versions, "best" is very hard to determine! In some cases, the SR approach may be picking up truly orthologous portions of the TD, but we think the simpler PanPipes perspective produces a much more understandable profile. In the end, perhaps this is where cyclic graph structures, though we have resisted them, are a more appropriate method (see Reviewer 2's comments, as well) .

6. Fig 7B. 'the lack of variants in the center of the profile indicates large structural differences...'. Were large structural differences

found? (since this is also mentioned in the discussion, it would be helpful to elaborate on it)

Please see related response to point 7 below.

7. Fig 7C. Please tell the reader in the text that Fig 7C is looking at the plateau region (and not the gap discussed in the prior sentence). What should the reader be seeing from Fig 7C? If I understand fig 7B correctly, the 'recombinants at the 5' end of the peak that are more tightly linked' are located at 25.5-25.7 Mb but 7C is considerably broader, 25.6-26.1. It would help the reader to annotate the relevant regions in both 7A and 7C so they know where to look, and to tell the reader why the broader region is shown in fig 7C.

Sorry, this figure reference was clearly misplaced; we are very grateful for this reviewer's excellent catch. We have corrected [ln 592-601] and added a supplemental figure confirming the cause of the visible variant gap (Figure S10) in a larger scope. Also, scale clarification has been added to figure 7 to avoid confusion. We did not include the dotplot for the entire mesa in the main figure because the NBS-LRR annotation get compressed and lost.

To answer the broader point, admittedly, the figure is somewhat open ended. This region bears the strongest association, but it also shows recombination suppression. There is slight evidence that the causal variant lies within the 5' portion of the "mesa", so it would be best to prioritize this region in future work. That said, the entire region is clearly a hotspot for resistance, probably of many kinds; such enrichment undermines our ability to strongly implicate that small linkage block within the large peak.

8. Discussion section (powdery mildew). "The strongest association lies in the first cluster of NBS-LRRs, in which both parents have 4 genes but MR1 has a 7kb insertion." This insertion was not mentioned in the results section, please mention and indicate where it is.

Added to results[ln 606-607] and hopefully clarified in conjunction with comment 7.

Other items:

1. Table 1. Are the assembly data in Table 1 from PacBio sequencing (and only PacBio)? Please clarify in the table title.

Added [ln 858].

2. Fig 1B and 1C are not mentioned in the text. They should be described and discussed.

See [ln 297-301].

3. “We confirmed that the community reference genome strain, DHL92, is also resistant”. Please inform the reader in this sentence that DHL92 does not have the 6.1 kb insertion.

See [ln 519-520].

5. Fig 7C. it would be helpful to tell the reader the number of NBS-LRR copies in each parent in the results section.

See [ln 602].

Reviewer #2 (Remarks to the Author):

The manuscript describes use of pangenome graph for analysis of bi-parental population and QTL discovery in melon. It also introduces a new pangenome construction and analysis tool/pipeline – PanPipes. I would like to start with the aspects of the manuscript I really enjoyed

- The manuscript is topical, the concept of pangenome graphs and their use as a reference is making its way into the plant genomic community
- The authors centre their analysis on application of pangenome graphs for analysis of bi-parental population sequenced using skim-Seq/GBS and the advantage of this approach over using a single community reference. Bi-parental populations are commonly used for QTL analysis in crop plants. The authors provide a clear, tangible example of a strong advantage of using pangenome graph and how it can practically improve genomic analyses
- The manuscript includes an in-depth discussion of three disease resistance/susceptibility loci. Each represent a different scenario of acquiring resistance/tolerance.

- The manuscript is very clearly written, making the explanations accessible even to the non-expert reader

Thanks for including positive aspects . . . this really helps us understand issues of most interest to the community.

My main major concern that it is hard to imagine that the PanPipes pipeline will be widely used by the community because it relies on a very old aligner, whose support has long been discontinued (<https://darlinglab.org/mauve/mauve.html>). There is no way of knowing how Mauve will perform on larger more repetitive genomes common in crop plants (soybean, wheat, barley; at least I don't know of any benchmarks which included it). On the GitHub page authors state 'Software, such as pggg, is a recent method to generate such graphs. In our hands, we have noticed that pggg often generates unwarranted cycles that violate the positional homology paradigm followed in PanPipes (see above). Our current preferred method is to rely on a multi-threaded implementation of the progressiveMauve algorithm, see GPA.' Which sadly makes the PanPipes already outdated.

I think PanPipes should be updated to at least allow passing of results from modern aligners. xmfa_tools allow conversion of Mauve xmfa file format into gfa, but probably the most common sequence alignment formats these days are maf/paf (I am including paf format here as in principle for bi-parental populations graph can be built with just two genomes and minimap2 and wfmash produce paf). There should at least be utility to convert maf and paf to xmfa and check if the gfa files (built by xmfa_tools using other aligners or other pipelines [pggg, cactus pangenome pipeline, minigraph]) meet the requirements to be used in PanPipes (and if they don't what needs to be done to get there).

We very much appreciate this major concern. It is one that we ourselves have, and it continues to be a discussion for the entire community (an excellent review recently published by Kille et al, 2022, "Multiple genome alignment in the telomere-to-telomere assembly era"). Though not described, we have evaluated numerous multi-chromosome alignment methods including Cactus and SibeliaZ variations, as well as those the reviewer mentioned. In our hands, we see little difference in accuracy, but progressiveMauve and pggg are the most scalable. True, progressiveMauve is old and no longer maintained, but it is still being actively developed for (see Henning and Nieselt, 2019, "Efficient Merging of

Genome Profile Alignments”). Also, its performance in complex alignments in the Alignathon challenge was on par with contemporary methods, and few methods have been developed since then. We have improved progressiveMauve’s default base-level alignment quality as well by parameterization sweeps and the addition of post-filters (discussed in <https://github.com/USDA-ARS-GBRU/PanPipes>).

Most importantly, progressiveMauve rigorously enforces linear alignments and full sequence inclusion, both of which were critical to us for proper interpretation and testing. The concept of including cycles (or loops) in the graph to represent paralogy or tandem duplication is a noble goal but, as it stands, we think linear enforcement serves genetic analysis better, although see comments the reviewer 1 (point 5) above.

We also want to emphasize that xmfa_tools is only one component of the PanPipes method. Any aligner that produces a GFA file (or format that can be converted to gfa) could be used by the downstream modules. We also note that, as we understand it, vgtools can directly take an MAF files, circumventing the need to import through xmfa_tools.

Still, we would like to leverage the effort we have put into xmfa_tools and allow users to “flatten” sequences in MAF format based on the hierarchical sorting algorithm we have implemented there. This would go beyond being a simple conversion script and address the complex task of linearizing cycles to their most appropriate position. We feel this is a standalone work though, and kindly request that this reviewer accept that we are working toward a tool that achieves (and more) the goal they describe.

Other, smaller concerns:

We hypothesize that this extensive divergence, in conjunction with natural selection, may have resulted from error prone repair often observed at tandem duplication boundaries.

This statement needs a reference.

See [ln 558].

Reviewer #3 (Remarks to the Author):

The authors sequenced and assembled two lines of melons, MR-1 (multi-disease resistant) and AY (susceptible), using long reads. The authors genotyped a RIL population of N=149 and did the genome-wide association analysis of the disease resistance using both the conventional single-reference approach and the pangenome approach. During the analysis, they found several bioinformatic

issues related to (I believe) the lack of mature ecosystem of pangenome analysis. To address (at least) some of the issues they found, the authors developed a new pipeline, PanPipes, that works with shallow coverage sequencing data better than existing methods. The authors analysis on disease resistance revealed that (putative) causal structural variations are more easily found with PanPipes and the pangenome approach.

The results suggest that the pangenome approach is a more powerful tool for agricultural and other non-model species.

It was a pleasure to read this paper. The paper brought me to a journey in which I see many bioinformatic issues the authors found during the analysis, which will shed light on challenges we, the genomics community, have to address in the near future. Not only the authors propose solutions to some of these issues, but also, they spur future development of pangenome analysis tools. More specifically, the graph-genome aligner challenge (ignoring too repetitive seeds, ignoring hypervariable regions for avoiding combinatorial explosion), the DAG challenge (tandem duplication has to be represented as an insertion because loops cannot be handled by the current tools, etc.), and the gene annotation challenge (how to annotate genes more accurately without prior knowledge of repetitive elements in the genome). I am not sure if the authors want to sell these, but however, I believe that this paper is going to be a landmark paper that will be cited dozens of times (or maybe hundreds) by the tool developer community.

Thank you; we greatly appreciate the positive feedback and hope you share this manuscript with fellow genomicists/geneticists. We certainly do not think we have solved all (or even most!) of the problems, but we have manually scrutinized hundreds of contrasting variant/allele calls at the base-level, and it is clear that the graph-based approach simply works better when highly divergence genomes are involved. In the end though, regardless of analysis strategy, the availability of the high-quality genomes (and good alignments) is the greatest benefit.

Put aside pointing out these challenges, they authors presented a model case where PanPipes finds a causal region for binary trait,

using biparental cross, which itself is intriguing. The conclusions are well supported by experimental evidence where possible.

Here are some minor issues:

1. Gene annotation section in Materials and Methods seems to lack a citation to TSEBRA (Lars Gabriel, et al, BMC Bioinfo. 2021)

We regret the omission; please see [ln 156].

2. Graph-based genotyping section says
> Hyper-variable regions were defined as 100 bp windows (relative to linear MR1 sequence ... were masked in subsequent analysis. Adding a bit of rationale for this procedure would help readers understand why this is necessary, though I guessed that this was for avoiding false positive calls.

See [ln 186-187].

3. In Data and software availability section, the authors state that the software is licensed under Creative Commons without specifying a variant, though the Creative Commons license have several variants. Taken together with “We have released all relevant software free to *the academic public*” (emphasis by me) in Discussion, the statement would lead to misunderstanding such as PanPipes is licensed under CC NY-NC-SA, which is a more restrictive license than CC0, which is indicated in the GitHub repository. My suggesting is to specify CC0

Thank you for the careful reading and advise; see [ln 748].

4. Fig 1C must indicate a region of Ns in some way (my suggestion is to show it as a gray rectangle in the dotplot). If I understand correctly, the dotplot in the middle must have a region of Ns, which appears like insertions in DHL92, which confused me for a few minutes.

Please see updated figure; these are place at the base of the DHL92 figure.

5. The second paragraph of the result section says “We observed that conventional repeat modeler approaches...,” though it was unclear to me what the “repeat modeler approaches” are. Is that an approach that we find repeat sequences by RepeatModeler2 (Jullien M Flynn, et al, PNAS, 2020) or EDTA (Shujun Ou, et al, Genome Biol, 2019)?

Good point and sorry for the loose statement; see [ln 269-270].

The following are just my comments, not issues.

The fact that the quantity of NBS-LRRs is not correlated with the disease resistance was intriguing because it underscores the importance of graph genome analysis. Conventional approaches such as presence-absence association analysis would not be able to reveal such an association. Also, Fig 6 & 7 are impressive.

Yes, based on this, we have become a bit suspicious of gene counting as an inference method in the absence of genetic evidence.

The paper convinced me that every reference genome should be reassembled from long reads. Given the timeframe of this study, it might have been difficult to reanalyze everything using the new reference genome by long-read sequencing [33]. However, this is not asking the authors to reanalyze, because I believe such an analysis would not affect the final conclusion very much.

Yes, that sequence was much improved PacBio, but, as mentioned, even more than timeframe issues, we wanted to consider cases where researchers might still be using a short-read or Sanger reference. Ideally, we could have assembled our own DHL92 using PacBio HiFi reads to have investigated the divergence versus quality question more fully. But, for this work, our main focus was on single reference versus graph-based and MR1/AY sequencing was sufficient for that purpose. We are discussing the possibility of work with the DHL92 group on integrating these efforts, as you suggest.

Reviewers' Comments:

Reviewer #1:

Remarks to the Author:

The authors have addressed my questions.

Reviewer #2:

Remarks to the Author:

The authors satisfactorily addressed my queries.

Pangenome graph construction relies on whole genome alignments which are unfortunately very challenging for many plant genomes. While I still think that there is no evidence that progressiveMauve approach will transfer well to other species, work presented here is an important stepping stone. I am looking forward to seeing (and applying) the outcomes of the authors' work on 'allowing users to "flatten" sequences in MAF format based on the hierarchical sorting algorithm we have implemented'.

I also believe that queries of Reviewer 3 were addressed in full. I strongly support publication of the manuscript

We greatly appreciate the reviewers who were able to confirm that we had addressed their comments.

All reviewers were satisfied with the revised manuscript and no further revisions were requested, as documented below:

REVIEWERS' COMMENTS

Reviewer #1 (Remarks to the Author):

The authors have addressed my questions.

Reviewer #2 (Remarks to the Author):

The authors satisfactorily addressed my queries.

Pangenome graph construction relies on whole genome alignments which are unfortunately very challenging for many plant genomes. While I still think that there is no evidence that progressiveMauve approach will transfer well to other species, work presented here is an important stepping stone. I am looking forward to seeing (and applying) the outcomes of the authors' work on 'allowing users to "flatten" sequences in MAF format based on the hierarchical sorting algorithm we have implemented'.

I also believe that queries of Reviewer 3 were addressed in full. I strongly support publication of the manuscript